# Online Measurement of Deposit Surface in Electron Beam Freeform Fabrication

**DOI:** 10.3390/s19184001

**Published:** 2019-09-16

**Authors:** Shuhe Chang, Haoyu Zhang, Haiying Xu, Xinghua Sang, Li Wang, Dong Du, Baohua Chang

**Affiliations:** 1Department of Mechanical Engineering, Tsinghua University, Beijing 100084, China; changsh15@mails.tsinghua.edu.cn (S.C.); z-hy18@mails.tsinghua.edu.cn (H.Z.); wanglidme@mail.tsinghua.edu.cn (L.W.); 2Key Laboratory for Advanced Materials Processing Technology, Ministry of Education, Tsinghua University, Beijing 100084, China; 3Manufacturing Technology Institute, Aviation Industry Corporation of China, Beijing 100024, China; xhyxhy@126.com (H.X.); sang_xh@163.com (X.S.)

**Keywords:** metal additive manufacturing, electron beam freeform fabrication, online monitoring, defect detection, 3D reconstruction

## Abstract

In the process of electron beam freeform fabrication (EBF3), due to the continuous change of thermal conditions and variability in wire feeding in the deposition process, geometric deviations are generated in the deposition of each layer. In order to prevent the layer-by-layer accumulation of the deviation, it is necessary to perform online geometry measurement for each deposition layer, based on which the error compensation can be done for the previous deposition layer in the next deposition layer. However, the traditional three-dimensional reconstruction method that employs structured laser cannot meet the requirements of long-term stable operation in the manufacturing process of EBF3. Therefore, this paper proposes a method to measure the deposit surfaces based on the position information of electron beam speckle, in which an electron beam is used to bombard the surface of the deposit to generate the speckle. Based on the structured information of the electron beam in the vacuum chamber, the three-dimensional reconstruction of the surface of the deposited parts is realized without need of additional structured laser sensor. In order to improve the detection accuracy, the detection error is theoretically analyzed and compensated. The absolute error after compensation is smaller than 0.1 mm, and the precision can reach 0.1%, which satisfies the requirements of 3D reconstruction of the deposited parts. An online measurement system is built for the surface of deposited parts in the process of electron beam freeform fabrication, which realizes the online 3D reconstruction of the surface of the deposited layer. In addition, in order to improve the detection stability of the whole system, the image processing algorithm suitable for this scene is designed. The reliability and speed of the algorithm are improved by ROI extraction, threshold segmentation, and expansion corrosion. In addition, the speckle size information can also reflect the thermal conditions of the surface of the deposited parts. Hence, it can be used for online detection of defects such as infusion and voids.

## 1. Introduction

Tiny defects in the manufacturing process of electron beam freeform fabrication can cause the failure of the whole deposited work piece, which is deposited over hundreds of hours. There is an urgent need for three-dimensional reconstruction of the surface for each deposit layer to detect defects and repair it in the next layer. In the past 20 years, metal additive manufacturing has been rapidly developed due to its near net shape advantage [1,2]. It mainly consists of two branches—powder bed and wire fed. Powder bed metal additive manufacturing includes Electron Beam Melting (EBM), Direct Metal Laser Sintering (DMLS), Selective Laser Melting (SLM), Selective Laser Sintering (SLS), and so on. The wire fed type includes electron beam freeform fabrication (EBF3), Hot Wire Gas Tungsten Arc Welding (HW-GTAW), Ion Fusion Formation (IFF), and so on. All metal additive manufacturing requires layer-by-layer deposition. Due to the instability of the heat input and the change of thermal conditions, the final deposited parts have various geometric errors compared to the digital model during the deposition process of each layer, such as the bulging at the beginning of the deposition due to material accumulation, the pits generated at the end of the deposition due to the electron beam impulse, and the collapse occurring at the corner due to the deterioration of the thermal conditions. Such geometric errors (especially the height error) in each layer can have a significant impact on the final part morphology and quality [3]. The thickness of each layer is very small in order to ensure the forming accuracy during the additive manufacturing process. A 10 cm high part usually has to be deposited over a hundred layers, so even a small error in each layer will cause the final part to fail after being accumulated layer by layer. This is particularly important in the electron beam freeform fabrication, which is often used to make large aerospace structural parts because of its high deposition efficiency [4,5]. The geometric deviations can easily produce defects, which result in the failure of parts deposited over hundreds of hours. This will greatly affect manufacturing efficiency. So, in the deposition process, each deposited layer needs to be reconstructed in three dimensions to obtain deposition deviation information. This deviation can then be compensated in the next layer of deposition process, so as to avoid part failure caused by error accumulation.

Because of the vacuum environment and strong metal vapor in the electron beam freeform fabrication process, the optical device will be quickly contaminated. This makes many existing 3D reconstruction techniques difficult to use in this environment, such as the structured laser technology which is widely used in 3D reconstruction of parts [6,7,8,9,10,11]. Dong et al. [12,13] realized the online measurement of helical rotor pitch by using structural laser technology. Sun et al. [14] achieved surface measurements on aero-engine blades based on laser triangulation. To solve for the problem that the line laser cannot stably obtain the bead information under the strong specular reflection condition, ZENG et al. [15,16,17] proposed a bead detection method using uniform illumination, directional light, and structure light. However, in the manufacturing process of electron beam freeform fabrication, due to the large amount of metal vapor, the lens in front of the laser is contaminated by a layer of metal vapor during long-term operation, so that the light transmittance is drastically reduced. In order to enable the laser speckle to be received by the CCD sensor, it is necessary to increase the laser power. Although the transmitted laser light intensity can meet the requirements of being detected by the CCD by increasing the laser power, most of the laser energy is absorbed by the lens. That can easily bring damage to the lens, and cannot meet the demand for long-term stable operation. Aubreton et al. [18] used a high-power laser to solve the problem that the laser could not work on the strongly specular surface. A thermal effect was generated on the surface of the workpiece to form a speckle. The height of the surface to be measured was obtained by extracting the position of the speckle. In EBF3, the surface of the workpiece can be directly bombarded by electron beam to generate a thermal effect, forming a speckle without an additional laser. This method can be used to achieve online measurement of the surface of the deposit during electron beam fusion deposition.

In this paper, a three-dimensional reconstruction system based on electron beam speckle is proposed for deposit surfaces, which uses the electron beam to bombard the deposit surfaces to produce thermal effect, and then form speckle. Based on the structural information of the electron beam in the vacuum chamber, three-dimensional reconstruction of the surface of the deposited parts is achieved. It overcomes the shortcomings of the laser specular reflection on the deposit surface and the contamination of optical lens by metal vapor.

## 2. Measurement Method of the Deposit Surface

### 2.1. Measurement of Deposit Surface Based On Electron Beam Speckle Position—Single Point

A schematic diagram of the online measurement method for the deposit surfaces in EBF3 is shown in Figure 1. After an electron beam is emitted from the cathode of the electron gun, it is accelerated by the anode to 1/3 to 1/2 of the speed of light. After being deflected by the electromagnetic deflection coils, the electron beam moves linearly to the surface of a deposit. Then it collides with the metal atoms, and generates a thermal effect. The localized heating of the deposited parts generates a speckle, and the position of the speckle in the camera sensor chip can reflect the height information of the surface of the deposit surface. For example, when the electron gun deflection coil current is zero (when the deflection coil is not zero, the electron beam is a diagonal line in space, and the situation is similar), the electron beam is perpendicularly incident on the surface of the deposit surface, and intersects the deposit surface at point A1, where a speckle is formed. The speckle light passes through the lens and is focused to point B1 of the CMOS sensor chip. When the surface height of the deposit surface rises to position 2, the intersection of the electron beam and the deposit surface is changed to A2, and the speckle is formed there. The speckle light passes through the lens and is focused to point B2 of the CMOS sensor chip. The height position of the deposit surface when the speckle is imaged at the CMOS center point is defined as *h*_1_. The height difference between the other position and the zero position is defined as △*h*. And the coordinates of the CMOS center point are defined as the zero point of the CMOS. The distance between other points on the CMOS and the zero point is defined as *h*’. The angle between the electron beam and the imaging optical axis is defined as α, the focal length of the lens is f. The object distance at the zero position of the deposit surface is *U*_1_, and the image distance is *V*_1_. Then, through the geometric calculation, the correspondence between the position of the speckle in the CMOS and the height position of the deposit surface is:(1)$$h={k*h\prime +h}_{1},$$
(2)$$k\;=\;\frac{U_{1}}{V_{1}*\sin\alpha },$$
where *h* is the height position of the object, *k* is the geometric magnification factor, *h*’ is the distance between other points on the CMOS and the zero point(B1), *h*_1_ is the zero height point of the object, *U*_1_ is the object distance at the zero position of object, *V*_1_ is the image distance at the zero position of object, and α is the angle between the electron beam and the imaging optical axis.

It can be seen from the Equation (1) that the position of the speckle in the CMOS sensor chip (*h*’) can reflect the real height of the deposit surface (*h*).

### 2.2. Measurement of Deposit Surface Based on Electron Beam Speckle Position—Whole Surface

The measurement method proposed in the previous section can be used to obtain the deposit surface height at a single point position, that is, the Z information in the three-dimensional coordinates. In order to realize the measurement of the whole deposit surface, the numerical control platform is moved to get all the points in the deposit surface. The moving path of the entire platform is shown in Figure 2. First, the Y-axis moves along a straight line at a constant speed while the X-axis is fixed. When the Y-axis scan completes for one X position, the X-axis is stepped to next position and the Y-axis continues to move in a straight line at the new X position. The scan of the entire deposit surface can be completed by repeating above steps.

In order to reconstruct the three-dimensional information of the entire deposit surface, it is necessary to synchronize the photos acquired by the camera and the X and Y coordinates when the platform is moving. In this way, the deposit surface height Z information extracted in each picture can be uniquely determined corresponding to a set of X and Y coordinates. The X and Y coordinate information of the platform is transmitted to the industrial computer through the OPC-UA communication module of the Siemens 840Dsl CNC system. The camera used for imaging transmits the acquired picture to the industrial computer through the network cable. The entire system is schematically shown in Figure 3. The system consists of an electron gun for generating an electron beam, a three-degree-of-freedom motion platform for placing the substrate, and an industrial camera for acquiring speckle image. The online measurement system is placed in a vacuum chamber, and the signal is passed through the vacuum chamber. Among them, the industrial computer is configured with an Intel 6 core E5-1650 processor, 3.5GHz operation frequency, and 32GB memory. The graphics card is Nvidia Quadro K2200. The camera is Image Source DMK 23GV024, the maximum frame rate is 115 fps, the pixel is 752 × 480, the CMOS sensor chip size is 1/3 inch, the pixel size is 6 μm × 6 μm, the numerical control system is Siemens 840Dsl, the electron gun power is 15 kW, the acceleration voltage is 60 kV. The beam current is 5 mA when scanning. The motion range of the platform is 1000 mm (X) × 500 mm (Y) × 500 mm (Z).

The workflow of the online measurement system is shown in Figure 4. The electron gun and the motion platform are controlled by Siemens 840Dsl CNC system. The electron beam is accelerated by a high voltage electric field to form a speckle on the deposit surface on the moving platform. At this time, the industrial computer generates a trigger signal which triggers the camera to capture images, and the OPC-UA client communicates with the host in the OPC-UA in the 840D numerical control system to obtain the current X and Y coordinates of the platform. The image captured by the camera is transmitted to the industrial computer through the GigE port, and the position of the current speckle in the CMOS is obtained through an image processing program. According to the formula in the previous section, the height information of the deposit surface, that is, the Z coordinate, is calculated. The X and Y coordinates acquired synchronously from the numerical control system are merged with the calculated Z coordinate, and the three-dimensional coordinates of the deposit surface at this position are obtained. Then the platform moves to the next position, at which time the industrial computer generates the trigger signal again, and the above process is repeated to obtain the three-dimensional information of the entire deposit surface.

### 2.3. Image Processing Algorithm to Extract the Position of the Speckle in the Camera Sensor Chip

As described in the previous section, in order to obtain the height of the deposit surface, it is necessary to know the position of the electron beam speckle in the CMOS sensor chip. The speckle image acquired by CMOS usually has noise, and the spatters generated during the scanning process also affect the calculation of the position of the speckle. This requires a fast and stable image processing algorithm. The entire image processing algorithm flow chart is shown in Figure 5.

First, the online measurement system acquires the speckle image through the camera, as shown in Figure 6a. Then the original image is binarized by a preset threshold to facilitate the subsequent extraction of the connected area. As shown in Figure 6b, after binarization, some surrounding noise with low gray value has been eliminated.

Since the electron beam propagates in a straight line in space, the image formed in the camera CMOS sensor is also a straight line, so the ROI (Region of interest) method can be used to remove surrounding noise and improve image processing efficiency. The image is ROI-limited using the ROI filter shown in Figure 6c. The processed speckle image is shown in Figure 6d, and it can be seen from the figure that the noise outside the ROI is eliminated.

The electron beam speckle is not a perfect circle. There are many stray speckles around the main beam speckle. The appearance of these stray speckles will bring extraction errors. These stray speckles are removed by a morphological processing algorithm that first erodes and then expands, as shown in Figure 6e.

The connected domain in the image is then extracted, and the largest area in the connected domain is identified as the speckle. By the center of gravity method, as shown in the Formula (3), the center coordinate Y of the electron beam speckle in the camera sensor chip and the speckle area are calculated as shown in Figure 6f.
(3)$$Y\;=\;\frac{m_{01}}{m_{00}}=\frac{\sum_{(i,j)\in S}jI(i,j)}{\sum_{(i,j)\in S}I(i,j)},$$

## 3. Error Analysis

After obtaining the central coordinate of the electron beam speckle in the camera sensor chip, that is, *h*’, it is necessary to know *k* and *h*_1_ in the Equation (1) in order to obtain the true height (*h*) of the deposit surface. Since it is very difficult to guarantee the precision when measuring the angle α (between the optical axis and the electron beam axis), the object distance *U*_1_, and the image distance *V*_1_, the calculation of *k* and *h*_1_ with measured α, *U*_1_, and *V*_1_ will introduce a large error to the system. Therefore, linear fitting method is used to estimate *k* and *h*_1_.

Firstly, the substrate is placed on the platform during the calibration process. The X and Y coordinates of the substrate are kept unchanged, and both the speckle image and the coordinate value of the coordinate axis Z (i.e., *h*) are acquired at the same time. After the electron beam speckle image is processed by the image processing algorithm, *h*’ is obtained. By moving the Z axis at a constant speed, a series of *h* and *h*’ can be obtained.

Figure 7 shows the results after linear fitting. We can see that *k* = 0.18198 and *h*_1_ = 119.7001. The correlation coefficient is 0.99987, which means that the linearity of the whole system is very high. Figure 8 is the fitted residual. It is found that there is a clear distribution of these errors, which will be analyzed later.

In order to improve the versatility of this camera, such as monitoring the shape of the molten pool or the transfer process of droplets during processing, a common optical lens is used in this system, which is coaxial with the camera. So, Shaman’s law is not satisfied, which requires that the intersection of the chip and the lens plane should be on the electron beam line. The speckle at any location can be focused on a point in the camera’s sensor chip when Shaman’s law is satisfied. However, the lens plane is parallel to the camera sensor chip plane in this system. These two planes intersect the electron beam linearly at two points *h*_1_’ and *h*_2_’, as shown in Figure 9.

Where *h*_1_′ can be expressed as:(4)$$h_{1}\prime \;=\;\frac{h*\sin\alpha {*V}_{1}}{U_{1}-h*\cos\alpha },$$

And *h*_2_′ can be expressed as:(5)$$h_{2}\prime \;=\;\frac{h*\sin\alpha *(V_{1}-f)}{f},$$

Then, the speckle center coordinate *h*’ can be obtained by:(6)$$h\prime \;=\;\frac{h_{1}\prime +h_{2}\prime }{2}$$

Figure 10 is the correspondence relationship between the calculated center (*h*’) of the electron beam speckle and real height (*h*) of the deposit surface. It can be seen that there is a relatively large nonlinearity error in the portion far from the center.

The nonlinearity error produced by the linear estimation in the formula can be seen from Figure 11. The absolute error at the edge of the CMOS chip is close to 1 mm with the measurement range of 120 mm. That means the relative accuracy is about 1%. After compensating for this nonlinear error, the error curve is shown in Figure 12. It can be seen that the error at the edge of the camera CMOS drops to 0.08 mm, the accuracy is improved to 0.1%, and the detection precision is significantly improved.

## 4. Results

A deposited part was placed on the platform in the vacuum chamber, which includes a 10-layer thin wall, a 1-layer single line and an 80 × 40 × 10 mm standard block, as shown in Figure 13. The platform in Figure 14 was moved according to the path shown in Figure 2. The three-dimensional reconstruction was performed according to the results calibrated in Figure 7. The results are shown in Figure 15. The shape of the deposit surface can be clearly discerned from the figure.

To verify the accuracy of the method in the actual surface, the region shown by the red dashed line in Figure 15 is selected. It was a flat surface. Therefore, the detected height value of this region is subtracted from the average value to obtain the detection error distribution of this region.

It can be seen from Figure 16 that the maximum error is less than 0.15 mm. The error distribution shows that the error in this region has obvious directionality. After outputting the error point cloud of the region, it can be found that there is an obvious tilt. This may be due to the unexpected gap between the standard block and the platform. The tilt error of the platform itself can also result in the measured upper surface of the standard block being not a horizontal plane. The surface was fitted using the least squares method to obtain a fitted surface, as shown in Figure 17.

The measured value is subtracted from the theoretical value at the fitted surface to obtain the flatness error of the detected area, as shown in Figure 18. It can be seen that the maximum error does not exceed 0.06 mm. It can meet the requirement of online 3D reconstruction of deposits during the electron beam fusion deposition additive manufacturing process.

## 5. Discussion

In the online measurement of deposit surface in EBF3, the speckle size information at each point can also be obtained, in addition to the position information of the speckle. The speckle size of each point is arranged in the scanning order to obtain an image, as shown in Figure 19.

It can be seen from the figure that the speckle size basically floats around 200, but there are three obvious sharp peaks in the middle, as shown by the red circles in Figure 19. The X and Y coordinates of the deposit surface are output to the X-axis and the Y-axis in Figure 20, and the speckle size at each point is output to the Z-axis. It can be easily found where the speckle size suddenly becomes larger. A corresponding point in the real deposit surface where the speckle suddenly becomes larger is shown in Figure 21. It can be seen that there is a spatter at this point. Since the spatter and the base substrate are only weakly bonded, the heat dissipation condition is worse there. When the electron beam moves to this position, the speckle becomes large because of the greater thermal effect. The detection of non-fusion defects on the surface of the deposits can be performed based on this effect.

## 6. Conclusions

In this paper, an online measurement system for deposit surface in EBF3 based on electron beam speckle position information is proposed. This method used electron beam to bombard the surface of the deposit to generate a speckle. Based on the structured information of the beam spot in the vacuum chamber, the 3D reconstruction of the surface of deposits was realized. No additional laser was required. In order to improve the detection accuracy of the method, the detection error was theoretically analyzed and compensated. The absolute error after compensation was smaller than 0.1 mm, and the precision can reach 0.1%, which satisfies the requirements of measurement of deposit surface. Based on this method, real-time OPCUA communication with Siemens 840Dsl numerical control system was carried out, and an online three-dimensional reconstruction system for deposition of EBF3 was established. In addition, in order to improve the detection stability of the whole system, the image processing algorithm was developed. The reliability and speed of the algorithm were improved by ROI extraction, threshold segmentation, and expansion corrosion.

## Figures and Tables

**Figure 1 sensors-19-04001-f001:**
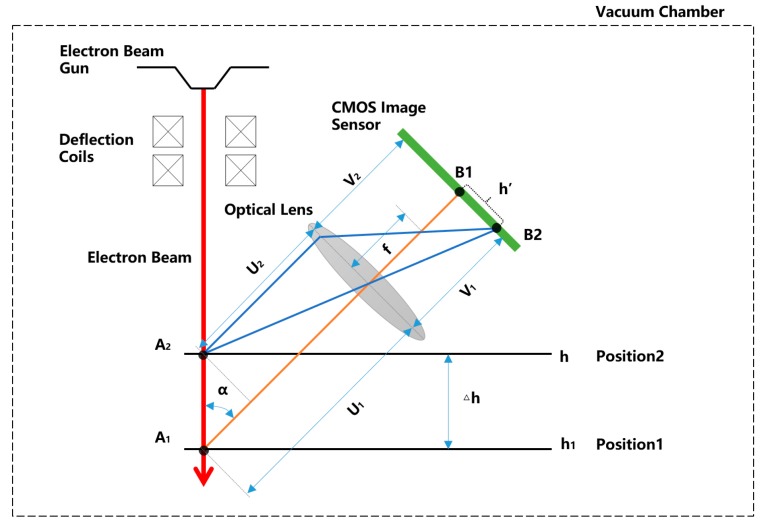
Schematic diagram of online measurement method for deposit surface in electron beam freeform fabrication (EBF3).

**Figure 2 sensors-19-04001-f002:**
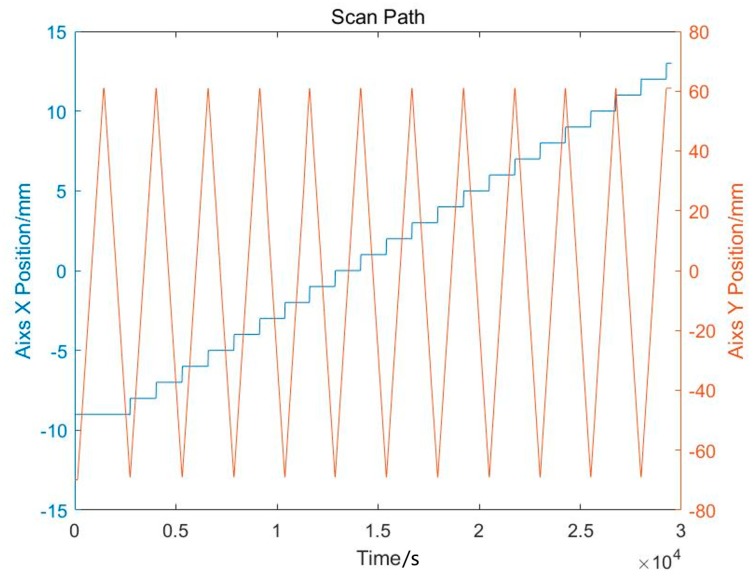
Moving path of the platform in X/Y plane.

**Figure 3 sensors-19-04001-f003:**
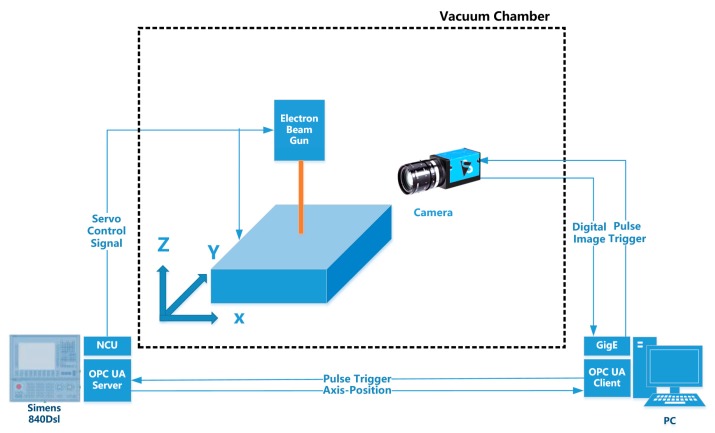
Composition of online measurement system based on electron beam speckle.

**Figure 4 sensors-19-04001-f004:**
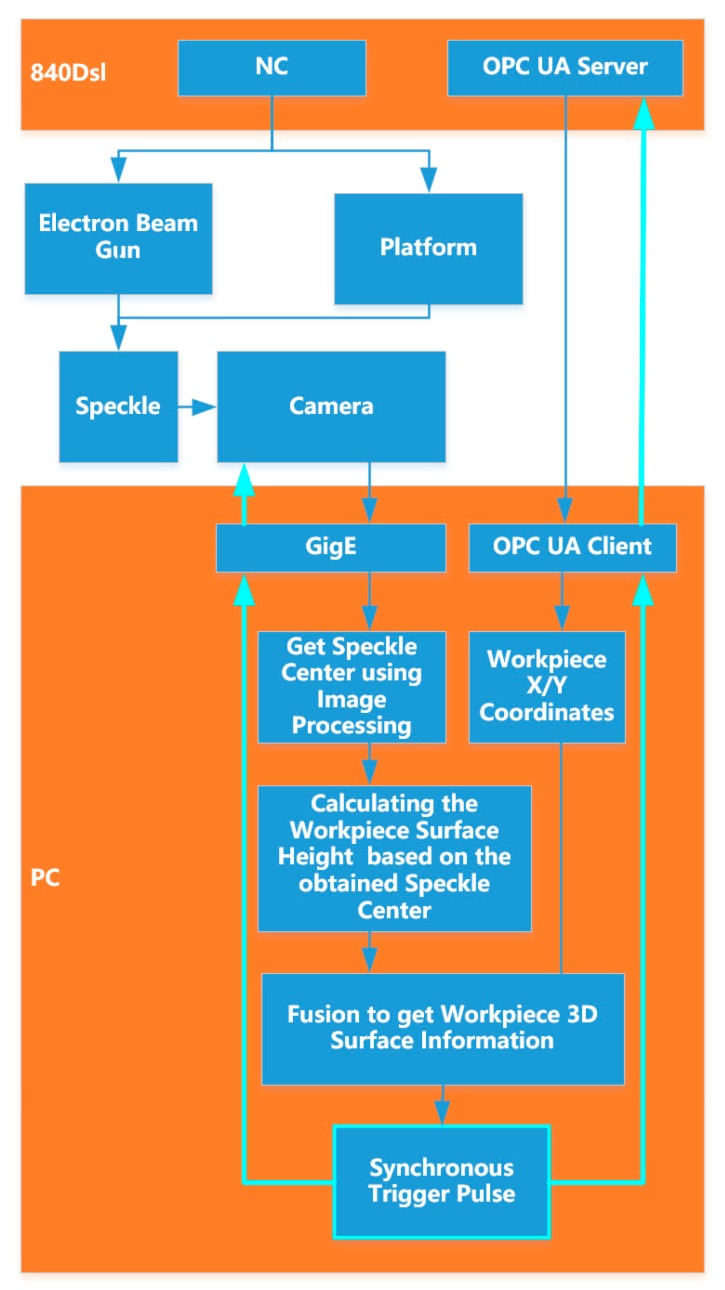
Flow chart of the measurement system.

**Figure 5 sensors-19-04001-f005:**
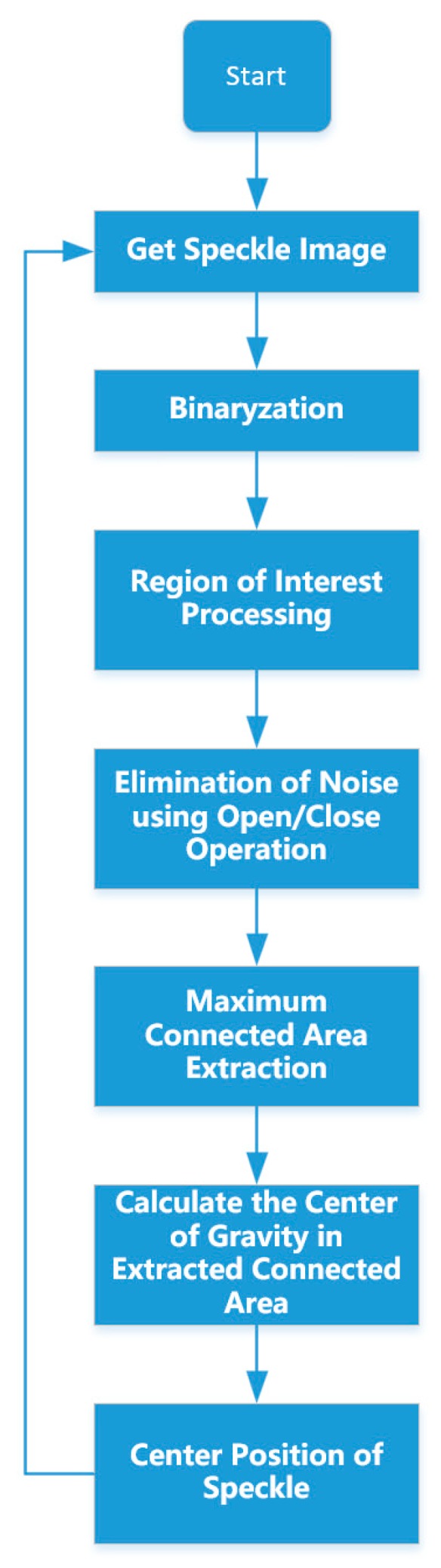
Flow chart of image processing algorithm.

**Figure 6 sensors-19-04001-f006:**
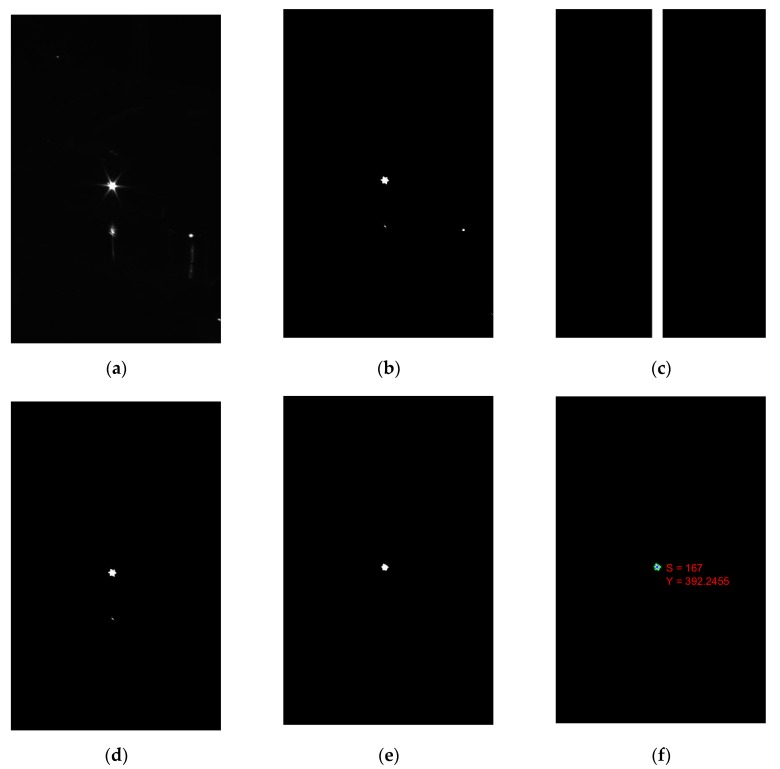
Image sequence (**a**) original image; (**b**) binarized image; (**c**) region of interest (ROI) filter; (**d**) image after ROI-limited; (**e**) image after corrosion and expansion; (**f**) extracted Y position and speckle size S.

**Figure 7 sensors-19-04001-f007:**
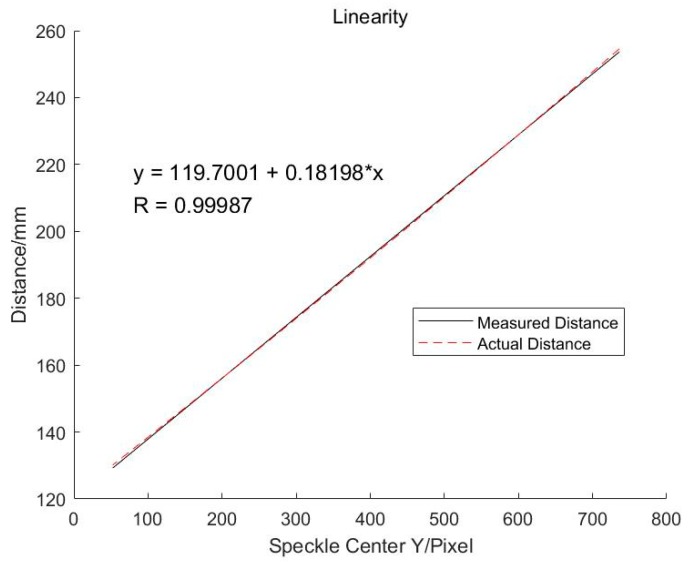
Linearity of the online measurement system.

**Figure 8 sensors-19-04001-f008:**
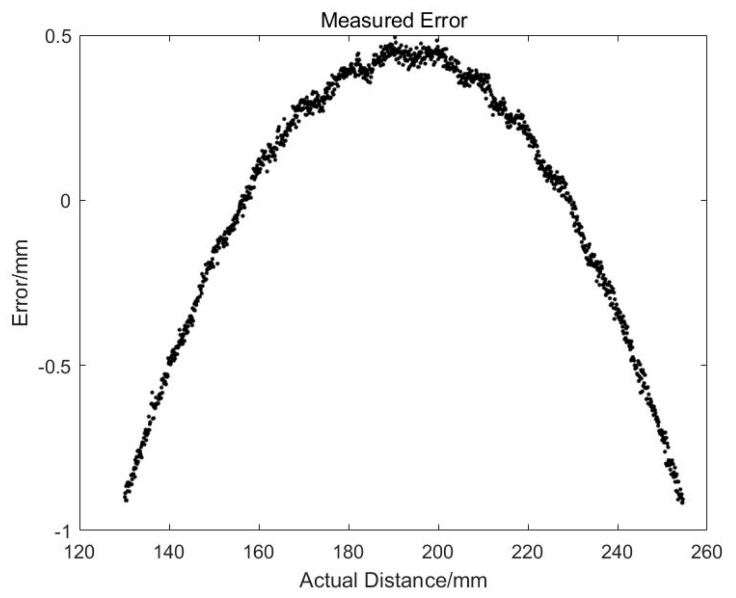
Errors of the online measurement system.

**Figure 9 sensors-19-04001-f009:**
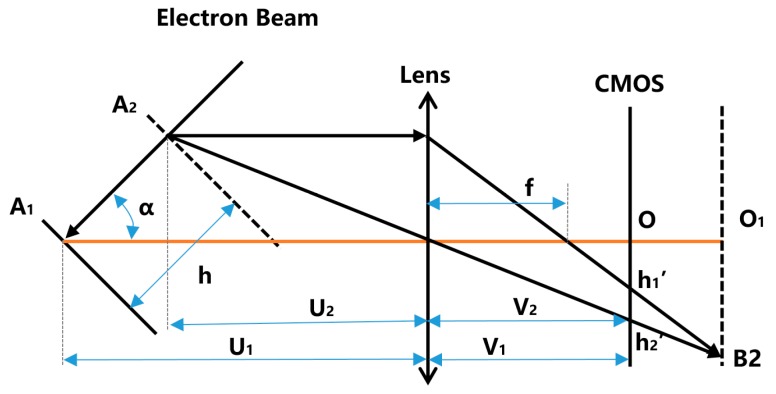
Error analysis schematic diagram.

**Figure 10 sensors-19-04001-f010:**
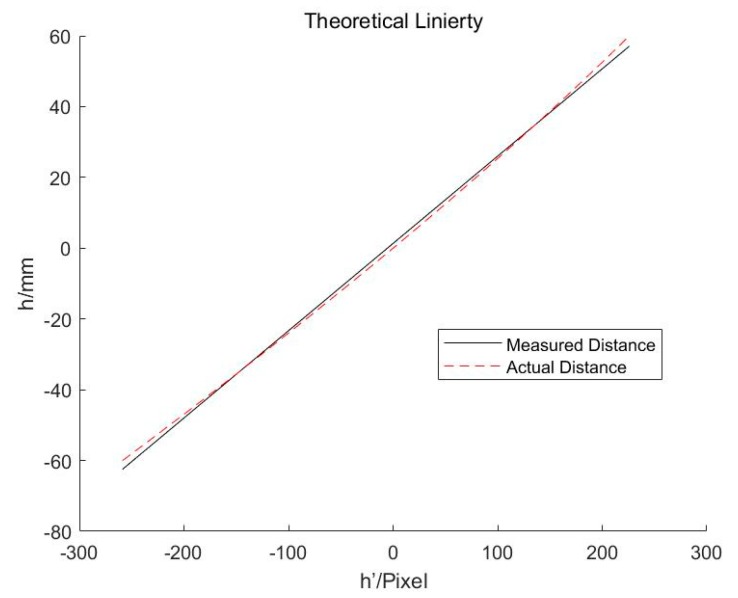
Theoretical linearity of the measurement system.

**Figure 11 sensors-19-04001-f011:**
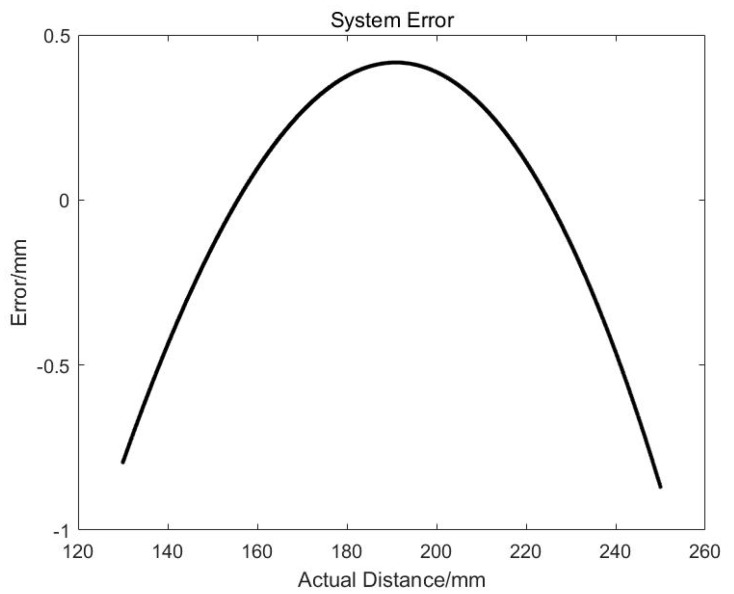
Theoretical errors of the measurement system.

**Figure 12 sensors-19-04001-f012:**
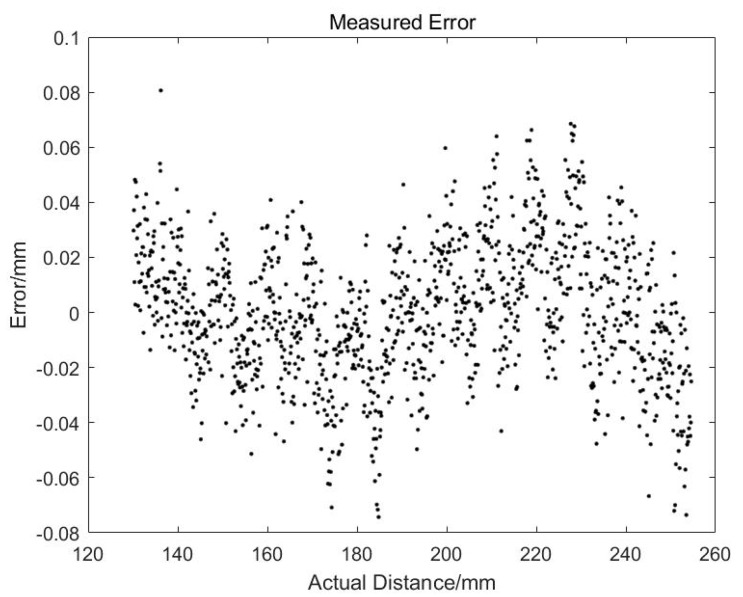
System error after compensating.

**Figure 13 sensors-19-04001-f013:**
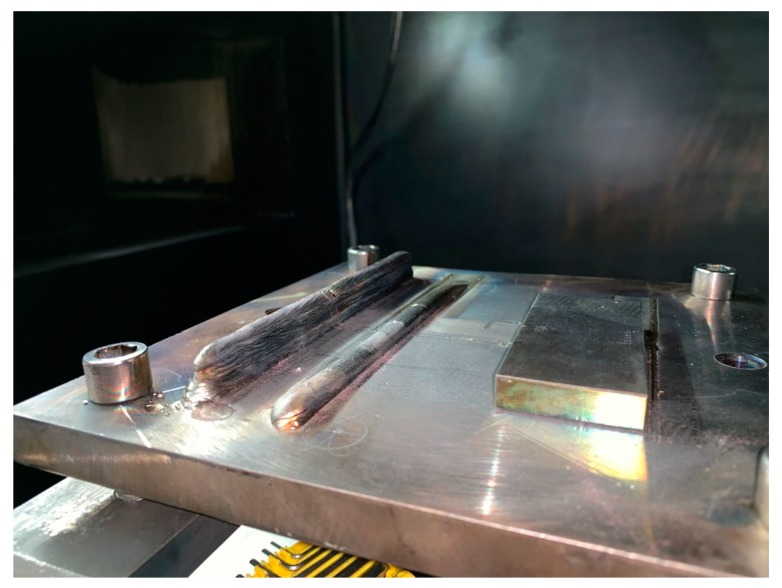
Deposited part to be measured.

**Figure 14 sensors-19-04001-f014:**
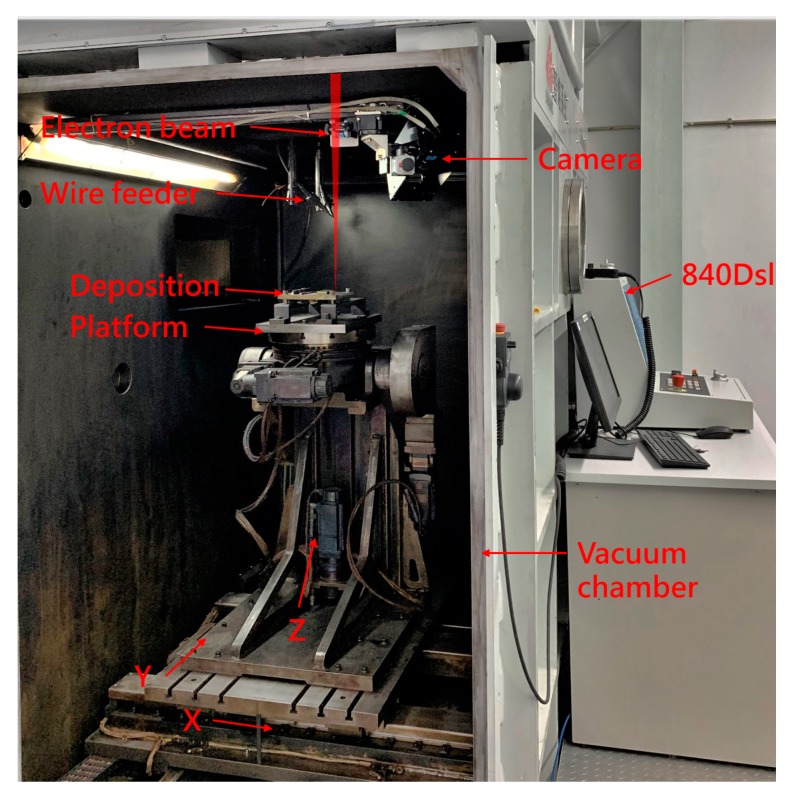
Configuration of the online measurement system.

**Figure 15 sensors-19-04001-f015:**
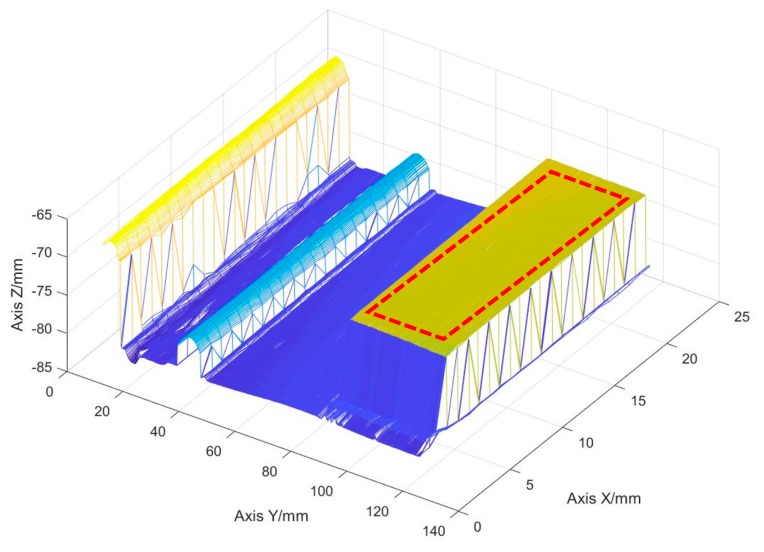
Measurement result of the deposit surface.

**Figure 16 sensors-19-04001-f016:**
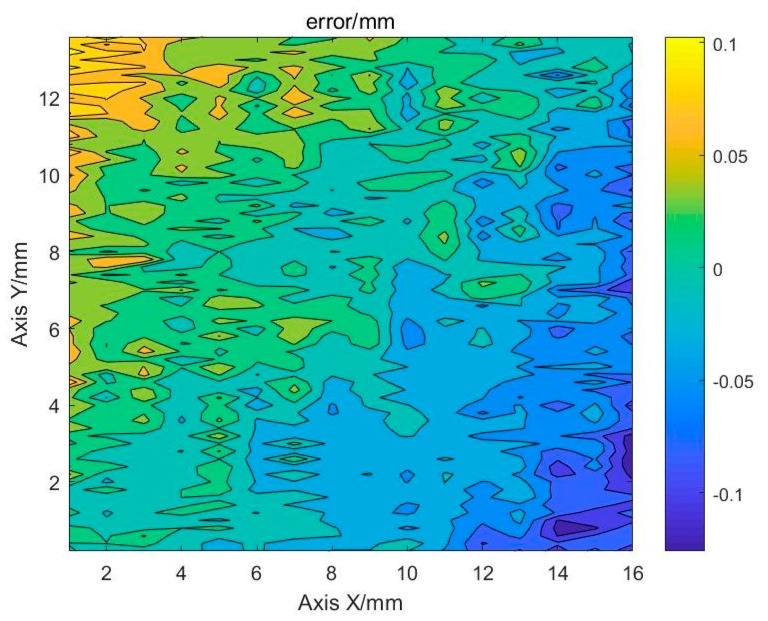
Flatness error of the standard block upper surface.

**Figure 17 sensors-19-04001-f017:**
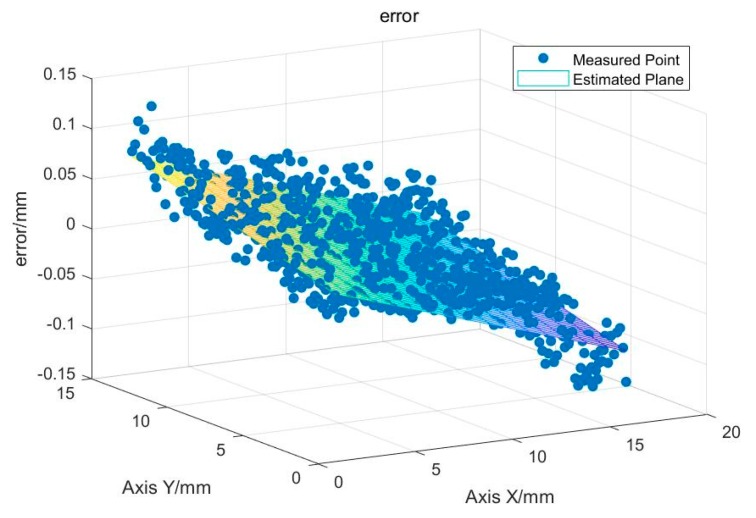
Fitting plane of the standard block upper surface.

**Figure 18 sensors-19-04001-f018:**
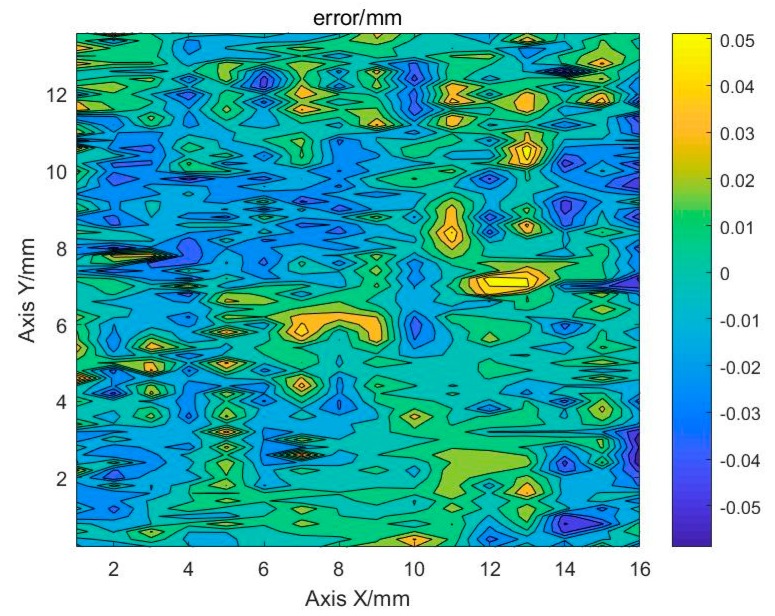
Flatness error after compensation of tilt.

**Figure 19 sensors-19-04001-f019:**
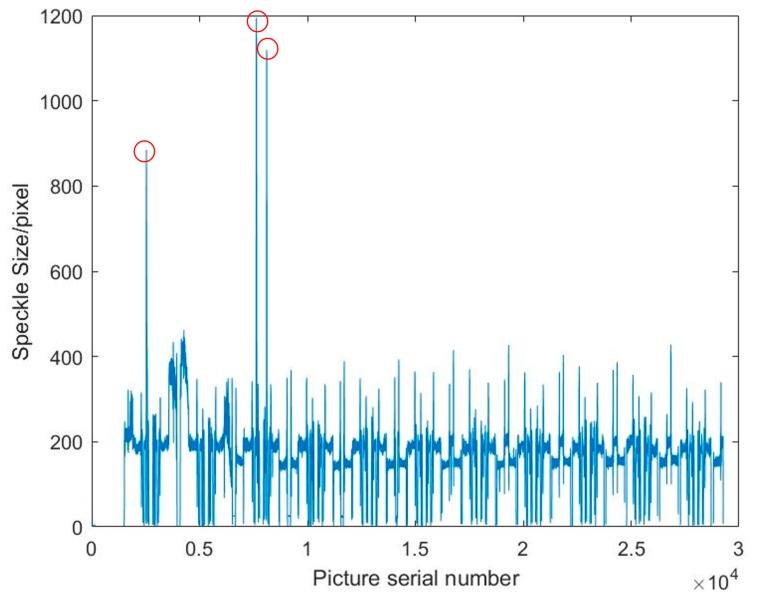
Curve of speckle size in measurement of the deposit surface.

**Figure 20 sensors-19-04001-f020:**
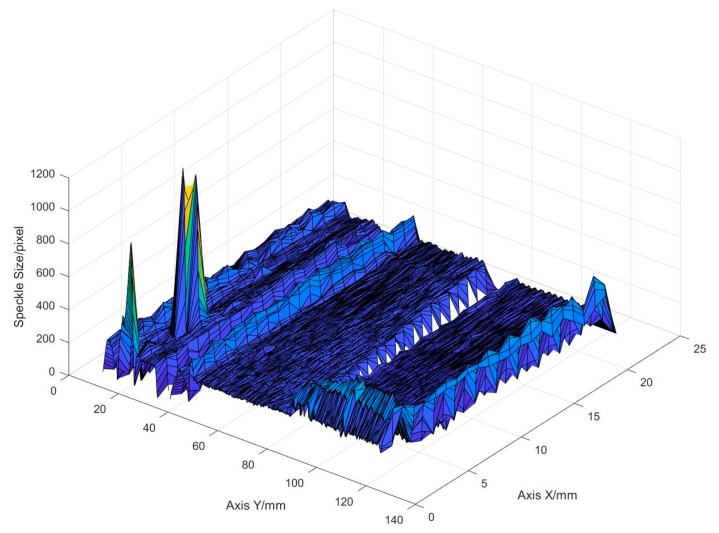
2D distribution of the speckle size peaks.

**Figure 21 sensors-19-04001-f021:**
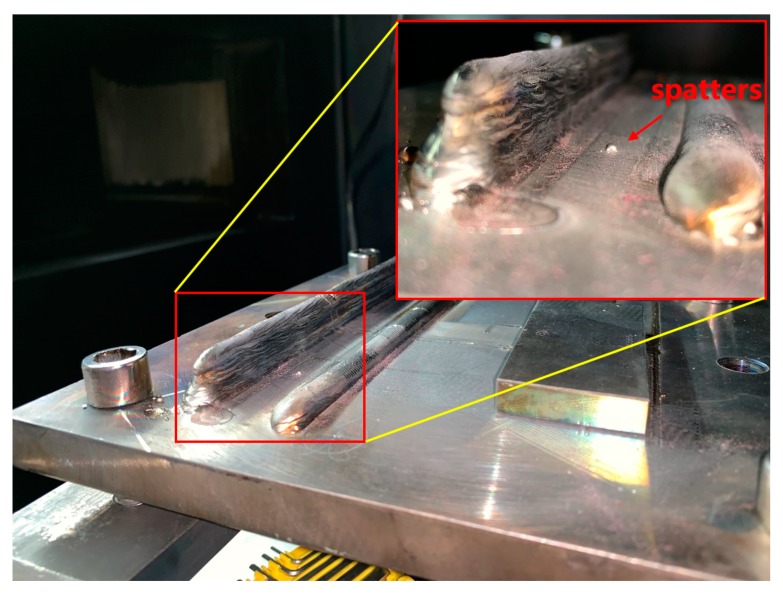
Spatter on the deposit surface.

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
