# Peer review of "Online Measurement of Deposit Surface in Electron Beam Freeform Fabrication"

_sensors, 2019, doi:10.3390/s19184001_

Round 1

Reviewer 1 Report

I would like to know that it is possible to detect the electron beam by a CMOS image sensor. or not. In my sense, when the electron beam is exposed to the surface of an object, the back scattered electrons and secondary electrons are generated. Which electrons did you used. If you used the back scattered electrons, these electron s are spattered all directions in the field. I think it is impossible to calculate the hight position of the object by geometric calculation as Figure 1. Would you please explain h(italic), k* in eq.(1) and l*, u(italic), v(italic and l in eq.(2)? I cannot find any word in Figure 1. In Figure1, is the lens an optical lens or an electrical lens or a Magnetic lens? In Figure 2, would you please indicate the unit of x-axis? In Figure 6, what is “S”?

Author Response

Response to Reviewer 1 Comments

Dear reviewer,

We are very grateful to your comments on out manuscript. According to your suggestion, corresponding revisions have been carried out to the manuscript, which are detailed as follows.

Point 1: I would like to know that it is possible to detect the electron beam by a CMOS image sensor. or not. In my sense, when the electron beam is exposed to the surface of an object, the back scattered electrons and secondary electrons are generated. Which electrons did you used. If you used the back scattered electrons, these electron s are spattered all directions in the field. I think it is impossible to calculate the height position of the object by geometric calculation as Figure 1. Would you please explain h(italic), k* in eq.(1) and l*, u(italic), v(italic and l in eq.(2)? I cannot find any word in Figure 1.

Response 1: Thank you for your suggestion. In fact, we don’t detect electrons. When the electron beam impinges on the surface of an object, there is a light speckle generated by thermal effect. We use the position information of the light speckle(captured by a CMOS camera) to get the height position of the object by geometric calculation as Figure 1, where h is the height position of the object, k is the geometric magnification factor which can be calculated by eq.(2), where h’ is the distance between other points on the CMOS and the zero point(B1), h1 is the zero height point of the object, U1 is the object distance at the zero position of object, V1 is the image distance at the zero position of object, α is the angle between the electron beam and the imaging optical axis. The letter l is a spelling mistake.  It's actually a corner 1 in V1 and U1.

Point 2: In Figure1, is the lens an optical lens or an electrical lens or a Magnetic lens? In Figure 2, would you please indicate the unit of x-axis? In Figure 6, what is “S”?

Response 2: The lens in Figure1 is an optical lens. And the unit of x-axis in Figure 2 is second. In Figure 6, S is the area size of the light speckle.

Thank you again for your suggestion. We have already revised the pictures and the annotations.

Reviewer 2 Report

This is a very clearly written paper on an interesting and important topic. This reviewer has only a couple of comments that should be clarified by the authors. The authors state a number of times that the speckle required for measurement of the height Δh, is generated by the electron beam through a "thermal effect." However, the authors never clearly explain the nature of this thermal effect. This reviewer is left with the impression that the thermal effect causes a localized thermal strain due to the electron beam spot that results in a localized "bump" on the substrate surface. Is this correct, or is the speckle formation due to some other thermal mechanism? Please explain the physics behind the speckle formation. If the electron beam creates a speckle due to the localized heating by the electron beam, then why is the speckle "not a perfect circle" as stated by authors on p. 6, line 184? Is the noncircular shape simply caused by the angle α shown in Fig. 1, or is there some other reason for a noncircular shape? Also, do the authors know why "there are many stray speckles around the main beam speckle"? (line 184). This is puzzling. Finally, if the electron beam thermal effect causes the speckle formation, won't thermal diffusion from this localized spot heating cause a drift in the height measurements at other points adjacent to the first spot as the heat diffuses (thermal diffusivity)? I believe that an explanation of how the electron beam creates the speckle and the true nature of the "thermal effect" will strengthen what is already a very nice paper!

Author Response

Response to Reviewer 2 Comments

Dear reviewer,

We are very grateful to your comments on out manuscript. According to your suggestion, corresponding revisions have been carried out to the manuscript, which are detailed as follows.

Point 1: This is a very clearly written paper on an interesting and important topic. This reviewer has only a couple of comments that should be clarified by the authors. The authors state a number of times that the speckle required for measurement of the height Δh, is generated by the electron beam through a "thermal effect." However, the authors never clearly explain the nature of this thermal effect. This reviewer is left with the impression that the thermal effect causes a localized thermal strain due to the electron beam spot that results in a localized "bump" on the substrate surface. Is this correct, or is the speckle formation due to some other thermal mechanism? Please explain the physics behind the speckle formation.

Response 1: Thank you for your suggestion. In fact, the speckle we observed is optical speckle instead of a localized "bump". When the electron beam bombards a point on the substrate, the kinetic energy of most electron beams converts to thermal energy which makes the temperature at this point on the substrate to rise. According to Planck's law, we see the point brighter than other areas through the optical camera. So we can get a light speckle on the image.

Point 2: If the electron beam creates a speckle due to the localized heating by the electron beam, then why is the speckle "not a perfect circle" as stated by authors on p. 6, line 184? Is the noncircular shape simply caused by the angle α shown in Fig. 1, or is there some other reason for a noncircular shape? Also, do the authors know why "there are many stray speckles around the main beam speckle"? (line 184). This is puzzling.

Response 2: Stray speckles and main beam speckle "not perfect circle" are mainly because of the bad quality of the electron beam. Astigmatism occurs when the electrons in the primary beam are exposed to a non-uniform magnetic field as they spiral round the optic axis. Astigmatism has several causes. It arises because the soft iron pole pieces comprising the electromagnetic lens cannot be fabricated with perfect cylindrical symmetry. The soft iron may also have micro-structural in homogeneities which cause local variations in the magnetic field strength. And the tungsten cathode of the electron beam gun begins to become imperfect with the burning loss in the process of processing.

Point 3: Finally, if the electron beam thermal effect causes the speckle formation, won't thermal diffusion from this localized spot heating cause a drift in the height measurements at other points adjacent to the first spot as the heat diffuses (thermal diffusivity)? I believe that an explanation of how the electron beam creates the speckle and the true nature of the "thermal effect" will strengthen what is already a very nice paper!

Response 3: Drift does happen when we increase the electron beam current from 5mA to 20mA. When we move to the next spot, the heat of last spot couldn't dissipate in time because of too much heat input. The shape of the light speckle we get changed from a circle to a line. So the drift errors becomes bigger when we use larger beam current. The brightness of the light speckle is big enough for extraction of the position information in a beam current of 5mA. And the heat input at a beam current of this level is quite small. The drift error is smaller than 0.1mm, which is accurate enough for measurement of the object surface. In the future, we will study this phenomenon more deeply in order to adapt this method to the wider application.

Thank you again for your suggestion. We have already revised the pictures and the annotations.
